# Research on Graphene and Its Derivatives in Oral Disease Treatment

**DOI:** 10.3390/ijms23094737

**Published:** 2022-04-25

**Authors:** Chengcheng Liu, Dan Tan, Xiaoli Chen, Jinfeng Liao, Leng Wu

**Affiliations:** 1State Key Laboratory of Oral Diseases, National Clinical Research Center for Oral Diseases, Department of Periodontics, West China School & Hospital of Stomatology, Sichuan University, Chengdu 610041, China; liuchengcheng519@163.com (C.L.); cxiaoli2022@163.com (X.C.); 2Department of Periodontics and Oral Mucosal Diseases, The Affiliated Stomatological Hospital of Southwest Medical University, Luzhou 646000, China; tandanpersist@163.com; 3State Key Laboratory of Oral Diseases, West China School & Hospital of Stomatology, Sichuan University, Chengdu 610041, China; 4Department of Stomatology, Tongji Hospital, Tongji Medical College, Huazhong University of Science and Technology, Wuhan 430030, China; 5School of Stomatology, Tongji Medical College, Huazhong University of Science and Technology, Wuhan 430030, China; 6Hubei Province Key Laboratory of Oral and Maxillofacial Development and Regeneration, Wuhan 430030, China

**Keywords:** graphene, graphene oxide, caries, pulp infection, periodontitis, osseointegration

## Abstract

Oral diseases present a global public health problem that imposes heavy financial burdens on individuals and health-care systems. Most oral health conditions can be treated in their early stage. Even if the early symptoms of oral diseases do not seem to cause significant discomfort, prompt treatment is essential for preventing their progression. Biomaterials with superior properties enable dental therapies with applications in restoration, therapeutic drug/protein delivery, and tissue regeneration. Graphene nanomaterials have many unique mechanical and physiochemical properties and can respond to the complex oral microenvironment, which includes oral microbiota colonization and high masticatory force. Research on graphene nanomaterials in dentistry, especially in caries, periodontitis therapy, and implant coatings, is progressing rapidly. Here, we review the development of graphene and its derivatives for dental disease therapy.

## 1. Introduction

Oral diseases constitute a public health problem worldwide with high prevalence (3.4 billion cases) and incidence (4.35 billion cases) [1]. Oral disorders can not only jeopardize local health but also affect general health. Dental caries and periodontitis are the two most common oral diseases. The former is a chronic infectious disease that destroys tooth hard tissues, with 4.24 billion incident cases in 2019 [2]. Periodontitis is a chronic infectious disease characterized by the progressive destruction of tooth-supporting tissue such as alveolar bone, even leading to tooth loss [3,4]. In 2019, periodontitis caused 7.09 million years lived with disability (YLD) globally, making it the seventh most prevalent disease globally, affecting 1.09 billion people globally [1]. Severe periodontitis affects 11.2% of the worldwide population [5]. Moreover, periodontitis is associated with many systematic diseases, such as diabetes, atherosclerosis, and Alzheimer’s disease, as demonstrated by numerous studies [6]. Thus, the effective treatment of oral diseases, especially periodontitis and dental caries, is of great importance.

The goal of dentistry is, through an achievable plan, to maintain or rehabilitate a person’s oral health, promoting general health. Filling, also called restoration, is the main treatment method for caries, which requires various materials, such as composite resins and adhesives. These materials are exposed to saliva, oral microbiota, high masticatory force, and abrasion, which can lead to failure of the treatment, the formation of secondary caries, fracture of the restoration, and microleakage caused by the shrinkage of the resin or the dissolution of adhesive [7,8]. Moreover, guided tissue regeneration (GTR) has been used in the repair of bone defects in many patients with severe periodontitis. However, there are still some problems with the current GTR materials, mainly including the stress-shielding effect, their inherent brittleness, the extremely low degradation rate of bioceramics, and poor antibacterial properties [9,10,11,12]. In addition, dental implants have become a conventional treatment for the substitution of missing teeth [13,14]. The current implant materials have shortcomings, such as poor bone healing and chronic infections, which limit their clinical application [15,16,17]. Titanium (Ti) is the most widely used implant material due to its biocompatibility [18]. However, it has been shown to generate alloy particles and ions into peri-implant tissues, resulting in bone loss and failure in osseointegration [19]. Therefore, to meet the clinical needs prevalent in dentistry, it is necessary to continuously develop novel materials for dental applications, such as functional dental restoration materials, dental composites with enhanced physicochemical and mechanical properties such as desired biocompatibility and adhesion to tooth tissues, and carriers for biological agent delivery.

Applications of nanotechnology have greatly contributed to the development of dentistry, and one of the more prominent aspects is the innovative development and application of nanomaterials in dental practice [20,21]. Graphene-based materials present outstanding characteristics, including remarkable mechanical properties, intrinsic antibacterial activity, very high surface area, good biological compatibility, and favorable differentiation of stem cells [22,23]. In the past two decades, graphene-based materials have demonstrated important applications in nanomedicine and nanobiotechnology, including tissue engineering, implants, antibacterial materials, drug delivery carriers, photothermal and photodynamic therapies, and biosensors, most of which are closely related to dentistry [24]. Therefore, graphene-based materials have broad application prospects in dentistry and have been extensively studied by numerous groups worldwide in the last few years. This paper focuses on the applications of graphene-based materials in oral disease treatment, particularly for dental caries and periodontitis. We review graphene and its derivatives regarding material basic composition, properties, fabrication, and compatibility. The role of graphene and its derivatives in improving the physical and chemical performance of dental materials and their application in dental carries, pulp and periapical diseases, periodontitis treatment, and dental implant restoration are also summarized, with emphasis on the inhibition of oral pathogens, facilitation of tissue regeneration, and the improvement of osseointegration. This provides a comprehensive overview of the research on graphene and its derivatives in oral disease treatment. The principal challenges and prospects were also discussed for the use of graphene-based materials in dental practice.

## 2. Graphene-Based Materials

### 2.1. Graphene and Its Derivatives

Graphene, the thinnest and strongest material known, is a single atomic sheet made up of sp^2^ (S, Px, Py)-hybridized carbon atoms with the arrangement of a honeycomb lattice [25,26]. It was successfully isolated by Andre Geim and Kostya Novoselov for the first time in 2004 [27]. Graphene has received much attention in the research of electronic and energy storage fields due to its excellent properties, including mechanical strength, modulus of elasticity and electronic properties, and different structures (e.g., graphene quantum dots, nanoribbons, and nanotubes) that can be easily fabricated [28,29,30,31,32]. In recent years, much research has focused on the application of graphene, the ‘superstar’ in materials, in biomedical fields such as tissue engineering, implants, antibacterial materials and biosensors, because of its unique two-dimensional form and outstanding physicochemical properties [33,34,35]. Numerous studies have established that the surface of graphene can be chemically functionalized with polymers, nanoparticles, and small molecules, which make graphene more suitable for drug delivery, cell and tumor imaging, and cancer photothermal therapy [27,28,36,37]. Graphene-based nanocomposites obtained by binding inorganic nanoparticles onto the surface of graphene have also been used for multimodal bioimaging and imaging-guided cancer therapy [38,39,40,41].

Graphene oxide (GO) and reduced graphene oxide (rGO) are two main graphene derivatives. GO was first discovered in 1859 and presented many oxygen-containing functionalities, such as hydroxyl epoxy and carboxyl groups, which contribute to the covalent or noncovalent combination of GO with biomolecules and other nanomaterials [42,43,44]. GO has more active sites than graphene while preserving the thin atom structure of graphene. Therefore, it is promising for use as a carrier for biomolecules and drugs, as well as for improving the bioactivity and mechanical performance of biomaterials. rGO is obtained by removing the oxygen functionalities of GO and recovering the conjugated structure, whereas a certain degree of oxygen-containing groups is found on the rGO surface. In contrast to GO, the oxygen-containing groups of rGO are much less abundant [45,46]. Several groups have demonstrated that GO is an effective carrier for the controlled delivery of substance P (SP) and bone morphogenetic protein 2 (BMP-2), resulting in improved osteointegration of dental Ti implants [47,48]. It has also been reported that both GO–Ti and rGO–Ti are good platforms for dexamethasone (DEX) loading, and DEX–GO–Ti showed a much higher potential to enhance cell proliferation, alkaline phosphatase (ALP) activities, and the expression of calcium and osteogenic differentiation-related genes than DEX–rGO–Ti and DEX-control [49].

### 2.2. Preparation of Graphene and Its Derivatives

Chemical vapor deposition, chemical-based methods, and micromechanical exfoliation of graphite are three main techniques used to synthesize graphene [50,51]. The graphene prepared through these methods exhibits excellent physical, chemical, and mechanical performance and can be transferred to various base materials [52]. The biomedical application of graphene and its derivatives may be limited to a certain extent due to the relatively high costs of industrial-scale pure graphene. Several methods have been developed to produce these high purity materials on the industrial scale at affordable prices [53]. For example, preparing graphene from low-cost carbon, the price of which is far less than that of raw graphite used in the exfoliation process, could save remarkable costs. “Flash Joule heating” is a unique method developed by Luong et al. that is able to instantly convert domestic waste carbon materials into high-purity crystalline graphene with a yield higher than 90%. This strategy could produce a product with a purity of over 99%, with no requirement of reactive gases, solvents, or furnaces, and as well as any purification step [54]. Therefore, this valuable, cost-effective, and sustainable technique leads to rapid progress in graphene research.

The oxidization of natural graphene results in graphite oxide, followed by sonication exfoliation to produce GO, the reduction of which forms rGO [53]. The chemical reduction of GO using green reducing agents has been proven eco-friendly, producing a highly dispersible and biocompatible product [55]. Recently, it has been of great interest to use natural reagents and environmentally friendly approaches to reduce GO. Many natural antioxidants have been used to reduce GO, such as amino acids, organic acids, and vitamins [56]. In addition, various plant extracts are used as reductants for GO due to the abundant polyphenols they contain. Polyphenol has a high tendency to oxidize and can react with the epoxide moiety to open the oxirane ring through bimolecular nucleophilic substitution. Similarly, the carbonyl and hydroxyl groups are subjected to nucleophilic attack by polyphenols while eliminating a water molecule. Through this reduction mechanism, GO can be successfully converted to rGO [57].

### 2.3. Compatibility of Graphene-Based Materials

Recently, graphene and its derivatives have given rise to great interest in the fields of biomedicine and dentistry. Similar to other inorganic nanomaterials, assessments of cytotoxicity are critical to their further clinical application. The biological toxicity of graphene and its derivatives has been systematically reviewed [58]. The toxicity of graphene, GO, and functionalized GO on different types of cells has been assessed, including fibroblasts, epithelial cells, and neuronal cells, using cell cultures with graphene-based materials [53,58,59]. Many studies have shown that uncoated GO or pristine graphene exhibits toxicity to various cell lines in a dose-dependent manner [60,61,62]. Furthermore, it seems that the production of reactive oxygen species (ROS), a critical characteristic of intracellular oxidative stress, as well as membrane damage and alterations in the expression of genes (e.g., Bcl-2, ERK, p38, JNK) related to apoptosis, are associated with the cytotoxicity of graphene-based materials in vitro [61,62,63,64]. Modification of the GO surface with hydrophobic macromolecules has been shown to result in a remarkable decrease in its cytotoxicity, including chitosan, Tween, artificial peroxidase, polyethylene glycol (PEG), dextran, and even proteins [36,65,66]. GO at a concentration of 50 g/mL has proven to be safe for most cell lines, and the concentration of rGO can be up to 60 g/mL [67]. Functionalization of rGO with polymer resulted in high water solubility and significant improvement of endothelial cell cytocompatibility, even at concentrations as high as 100 g/mL [65]. Studies have also assessed the oral toxicity of a few graphene-based materials, mainly GO, rGO, polymethyl methacrylate (PMMA) resin loaded with graphene-Ag nanoparticles (G-AgNps), and GO/rGO-incorporated sodium alginate (GOSA/rGOSA) scaffolds [68,69,70,71]. GO exhibited the lowest cytotoxic effect on human dental follicle stem cells (hDFSCs), inducing oxidative stress without damaging the cell membrane. Although showing a good safety profile at a concentration of 4 μg/mL, nitrogen-doped graphene at a high concentration (40 μg/mL) reduced the cell viability and caused membrane damage via mechanical effects. In contrast, thermally, rGO presented high cytotoxic effects. Thus, they suggested GO and nitrogen-doped graphene as valuable candidates for application in dental nanomaterials [69]. PMMA resin loaded with G-AgNp could decrease the viability of dysplastic oral keratinocytes and dental pulp stem cells (DPSCs), but the cell viability remained over 75% compared to controls [68]. Dreanca A et al. also investigated the biocompatibility and toxicity of two graphene composite dental materials, namely, a light-curing hybrid restorative composite and cement. In their study, there was no notable in vitro cytotoxicity on hDFSCs and dysplastic oral keratinocytes observed, and no in vivo symptoms of acute toxicity or local inflammation were found in the animals at 7 weeks after the implantation of these materials in a mandibular defect. These findings suggested the good biocompatibility of graphene dental composites used in dentistry [70]. However, many aspects of graphene and its derivatives cytotoxicity remain to be further investigated, such as how exactly the materials’ sizes and surface chemistry regulate the material–cell interactions and the molecular mechanism of toxicity. Moreover, toxicity testing of new compounds or materials in vivo exposure on experimental animals is essential for their clinical application. Several lines of evidence have suggested that fine-tuning the surface chemistry is critical for optimizing the pharmacokinetics and distribution of graphene in vivo for intended applications in biomedicine. Intravenous injection of uncoated GO has been reported to induce adverse effects, including strong aggregation of human platelets, pulmonary edema, high thrombogenicity, and granuloma formation in mice [72,73]. Moreover, functional groups (e.g., amine and carboxyl) and polymers (e.g., dextran, PEG, and chitosan) on graphene surfaces have been proven to reduce the toxicity of graphene in vivo [74]. Collectively, these results support the association between the in vivo toxicity of graphene-based materials and their surface coatings as well as the possibility for well-designed surface modifications of graphene-based materials to effectively reduce their toxicity.

## 3. Improving the Physical and Chemical Performance of Dental Materials

Decreasing the restoration failure caused by bulk or marginal fractures, as well as reducing the risk of secondary caries, to minimize the demand to replace restorations is a crucial goal of contemporary dentistry [75,76]. The composition of the dental material is a key factor influencing the lifespan of dental restorations [77]. Microleakage owing to the poor resistance and insufficient adhesion of dental materials to tooth tissues can lead to biofilm accumulation and consequent failure of restoration. A rate as high as 50–60% of secondary caries due to microleakage has been reported [78].

Graphene-based materials have been introduced into dentistry to improve the performance of dental materials. Evidence has indicated that graphene can enhance the mechanical and physicochemical properties of biomaterials, being biocompatible and noncytotoxic in the form of few-layer graphene (FLG), generally 1 to 6 layers. Therefore, FLG should be an ideal material that can be incorporated into dental polymers, thereby increasing their strength and durability. Malik S et al. [79] fabricated graphene dental-polymer composites by incorporating FLG into a common dental polymer. They found that the addition of graphene (0.2 wt%) increased the compressive strength and compressive modulus by 27% and 22%, respectively. Glass ionomer cements (GICs), as widely recognized popular restorative materials in dentistry, often result in restoration failures because of poor mechanical performance and secondary caries. Sun L et al. found that the mechanical blending of fluorinated graphene (FG) with traditional GIC powder improved the composites’ mechanical properties significantly, without adverse effects on the color and solubility, as well as fluoride ion releasing properties of the composites. Specifically, the Vickers microhardness and compressive strength were enhanced with increasing FG content, and the Vickers microhardness and compressive strength of the composites enriched with 2 wt% FG were increased by approximately 60% compared to the control GICs. The GIC/FG (4 wt%) composites showed a significant decrease in the volume wear rate and friction coefficient. Interestingly, the addition of FG into GICs could also enhance the solubility resistance of the composites with the exception of the content of 4 wt% [80]. However, findings obtained via incorporation of up to 2 wt% of reduced graphene–silver nanoparticles into conventional glass ionomer powder have suggested that nanocomposites could not affect the composite surface microhardness and flexural strength [81].

In addition to GICs, bioactive calcium silicate bone cement is widely used in dentistry for endodontic treatments such as the management of perforations, retrograde root filling, and pulp capping due to its strong sealing ability and mineralization inducing ability, but it still has some problems in clinical application, including a long setting time and unsatisfactory physico-mechanical properties. Dubey et al. [82] tried to modify two bioactive cements, Biodentine and Endocem Zr, using graphene nanosheets and found that the addition of graphene nanosheets (1 wt% and 3 wt%) resulted in shortened setting time and increased hardness for both materials. Moreover, Endocem Zr enriched with 1 wt% and 7 wt% graphene nanosheets and Biodentine enriched with 5 wt% graphene nanosheets presented higher mineralization than the controls. However, graphene nanosheets had a negative effect on the push-out strength of Endocem Zr [82]. Therefore, notwithstanding its potential to improve the physico-mechanical performance of bioactive cements, graphene nanosheets should be used with caution when effective bonding is needed. Resin-based dental adhesion has been successfully applied in minimally invasive operative dentistry, whereas the stability and durability of adhesive interfaces remain to be improved. Hybrid layer (HL) deterioration, especially collagen fibril degradation, has been regarded as the main reason for the failure of the resin–dentin bond interface. Chen W et al. [83] demonstrated that graphene quantum dots (GQDs) and carbodiimide can synergistically inhibit collagen fiber hydrolysis and enhance adhesion durability. Specifically, GQDs cross-linked collagen noncovalently and remarkably inhibited the activity of collagenase but with limited and unstable ability of the anti-enzymatic hydrolysis of collagen. When combined with carbodiimide, GQDs could covalently bond to collagen fibers, simultaneously improving the anti-enzymatic hydrolysis ability of collagen fibers and inhibiting collagenase activity. Moreover, under the acid-etched and rinse adhesive system, GQDs with 1-ethyl-3-(3-dimethylaminopropyl) carbodiimide hydrochloride improved the bonding strength after thermocycling, inhibited matrix metalloprotein activity in situ, and promoted HL integrity after thermocycling [83].

Unlike GICs and bioactive calcium silicate bone cement, PMMA, known as “organic glass”, is widely used as a denture material due to its superior heat resistance, high strength, and low cost. However, it suffers from brittleness and poor impact resistance. A previous study demonstrated that incorporation of G–AgNp improved the flexural strength of PMMA resin [68]. Furthermore, the loading of G–AgNp (1% and 2%) into PMMA resin significantly enhanced the compression behavior and tensile strength, as well as the flexural profile of the PMMA material. Compared to the control material, a content of 1% G–AgNp appeared to be sufficient for the PMMA resin to withstand higher applied loads and presented higher flexural and tensile strength. However, G–AgNp at a content of 2% showed lower water absorption, thereby reducing the risks of degradation effects mediated by water [84]. Some clinical data also support this point. Azevedo L et al. made a definitive maxillary prosthesis using GO-incorporated PMMA resin to improve the resin’s mechanical properties. They found that after 8 months of placing the prosthesis, no mechanical, aesthetic, or biologic complications were reported, with healthy and stable soft tissues. This suggested that GO-loaded PMMA resins appear to be applicable in prosthetic rehabilitation, but rigorous scientific support and the benefits of these new technologies and materials still need to be further explored [85]. In contrast, Paz E et al. [86] found that there is no substantial difference in thermal properties between PMMA bone cement and cement enriched with 0.1 wt% G or GO with respect to the extent of the polymerization reaction, heat generation, thermal conductivity, and glass transition temperature.

In addition, GO could improve the performance of adhesive and silane primers and protect Ti substrates [87,88,89]. Specifically, GO-enriched adhesive and the control adhesive had similar tooth dentin interactions, along with the formation of an HL. In the absence of nanoleakage, GO-enriched adhesive exhibits bond strength and durability comparable to those of resin-dentin bonds [87]. GO-modified experimental silane primers enhanced the shear bond strength between the resin composites and zirconia [88]. GO/chitosan/hydroxyapatite (GO/CS/HA) coatings can effectively protect Ti substrates from corrosion [89].

## 4. Potential Application of Graphene-Based Materials in Oral Disease Treatment

### 4.1. Inhibiting Cariogenic Bacteria and Preventing Demineralization of Teeth

Dental caries is a disease that relies on cariogenic biofilms, which are highly organized microbial communities embedded in a cohesive matrix of extracellular polymers, mainly extracellular polysaccharide (EPS) [90]. *Streptococcus mutans* is a key cariogenic pathogen and can synthesize insoluble EPS using dietary sucrose, thereby facilitating bacterial adhesion–cohesion and accumulation on the tooth surface, subsequently promoting the formation of caries-causing acidogenic biofilms [91]. Therefore, developing *S. mutans*-targeting materials is of great importance for controlling the pathologic condition. Although widely explored biocides, such as chlorhexidine, present a strong ability to inhibit *S. mutans*, they also suppress the growth of beneficial bacteria, disrupting oral microbiota homeostasis [92]. Currently, research is of great interest on nanomaterials such as graphene and its derivatives in the prevention and treatment of dental caries.

Abundant evidence has demonstrated that graphene and its derivatives are promising anti-caries nanomaterials due to their impressive ability to inhibit cariogenic bacteria and prevent tooth demineralization [81,93,94,95,96,97,98,99] (Figure 1). To summarize, graphene and GO may directly inhibit cariogenic bacteria and indirectly enhance the antibacterial properties of metallic nanomaterials, such as silver, copper, and zinc oxide nanoparticles [81,93,94,95,96,97,98,99]. It has been demonstrated that graphene nanoplatelets, GICs enriched with FG, GO nanoparticles, and GO nanosheets all can remarkably inhibit the adhesion and growth of *S. mutans* in vitro [80,93,94,96]. For instance, GICs enriched with FG significantly decreased the colony count against *S. mutans*, and the antibacterial properties improved with increasing FG content. When 4 wt% FG was added, the antibacterial rate of *S. mutans* reached 85.27%. Abundant evidence has demonstrated that graphene and its derivatives are promising anti-caries nanomaterials due to their impressive ability to inhibit cariogenic bacteria and prevent tooth demineralization [80]. A study reported that GO nanosheets were highly effective in suppressing the growth of *S. mutans* using the 3-(4,5-dimethylthiazol-2-yl)-2,5-diphenyl tetrazolium bromide (MTT) reduced test, colony forming unit counting, growth curve observation, and live/dead fluorescent staining [96]. Graphene-based metal nanomaterials are also regarded as potent agents against cariogenic pathogens. Chen J et al. [81] found that adding 2 wt% reduced graphene-silver nanoparticles into glass ions and significantly reduced the number of *S. mutans*. Consistent with this result, the addition of 0.25% graphene nanoplatelets doped with silver nanoparticles into adhesives showed optimized antibiofilm properties against *S. mutans* without affecting the standard adhesion characteristics regarding bond strength, leakage expression and durability at the resin–dentin interface [97]. In addition to inhibiting the growth of *S. mutans*, graphene oxide–copper nanocomposites (GO–Cu) and graphene/zinc oxide nanocomposites (GZNC) could also reduce the biomass of the *S. mutans* biofilm and suppress cariogenic properties of *S. mutans*, such as acid production and glucan formation [98,99]. The mechanisms underlying how graphene and its derivatives could inhibit cariogenic pathogens and biofilm formation mainly include causing (1) mechanical damage of the nanostructured materials on the bacterial cell wall and (2) altering the biofilm architecture and impairing EPS production and distribution [93,98,99,100].

Interestingly, there are studies that combine GO with new technologies to achieve its antibacterial effect (Figure 1). When ionically bonded to cationic polymers, GO can efficiently deliver nucleic acids, increasing the uptake process of genes and protecting nucleic acids from the lysosomal pathway [101,102]. Antisense vicR (AsvicR) RNA has been reported to reduce the transcription of virulence genes, thereby reducing biofilm formation in *S. mutans* [103]. Wu S et al. [104] developed a GO plasmid transformation system using interacting GO–polyethylenimine (PEI) complexes loaded with an AsvicR-expressing plasmid (GOPEI–AsvicR). They showed an efficient delivery of the AsvicR-expressing plasmid into *S. mutans* cells using GOPEI complexes, and a reduction in virulence-associated gene (gtfBCD and gbpB) expression due to AsvicR was observed. Both GO and AsvicR alone could significantly suppress biofilm formation and EPS production. In contrast, GOPEI–AsvicR exhibited much more remarkable inhibitory effects with respect to virulence-associated gene expression, biofilm aggregation, and EPS accumulation, which may be related to silencing the expression of the *vicR* gene and the physical effect of GO. They suggested preserving nanographene oxide with AsvicR RNA as a more effective strategy for dental caries management [104] (Figure 2). Recently, antimicrobial photodynamic therapy (aPDT) has emerged as an effective adjunctive therapy for intracanal microbiota, during which the type of photosensitizer plays a key role in the efficiency. Indocyanine green (ICG), a widely used anionic photosensitizer in therapeutic applications for dental caries, suffers from poor stability and concentration-dependent aggregation. It was shown that, as a novel nanocarrier, multifunctionalized GO significantly enhanced ICG loading and stability and improved the inhibitory effects of ICGs as photosensitizers in aPDT against *S. mutans* [105].

Demineralization is the first sign of dental caries, and remineralization repairs the outer layer of the teeth. Several studies have shown the involvement of graphene and its derivatives in inhibiting demineralization and promoting remineralization of enamel [106,107,108,109]. For instance, the mixture of bioactive glass and GO (3 wt% or 5 wt%) markedly increased the microhardness and dose-dependent anti-demineralization effect of Low-Viscosity Transbond XT, which is an adhesive, without significant influence on the shear bond strength, adhesive remnant index, and in vitro cytotoxicity [106]. Son S et al. [109] successfully synthesized a mesoporous bioactive glass and coated it with GO quantum dots, showing spherical nanoparticle formation and a uniform coating of GO quantum dots in the mesopores of mesoporous bioactive glass. Using ion release and in vitro mineralization tests, they revealed that GO quantum dot-coated mesoporous bioactive glass promoted the formation of hydroxyapatite rather than inhibiting the release of calcium, silicon, and phosphate ions [109]. However, the effect of graphene and its derivatives on the demineralization and remineralization of teeth requires further investigation in vivo.

### 4.2. Control of Dental Pulp Infection and Promotion of hDPSC Differentiation

Without effective control, caries can develop into pulp and periapical disease. Endodontic treatments, such as root canal treatment (RCT), pulp regeneration, and apical induction are the main and effective treatments for teeth with pulpitis or apical periodontitis. RCT is a fundamental step to remove infected tissue and pathogens in the root canal system. To reach the goals of RCT, filling materials used to occupy complex root canal systems are supposed to have the desired sealing ability, biocompatibility, and antibacterial properties. Persistent infection in confined areas of the root canal system is a main reason for root canal treatment failure and posttreatment apical periodontitis. Owing to the less accessible areas of the root canal system, mechanical debridement and chemical irrigation during RCT are often not enough to eliminate bacteria from the root canal system. Therefore, it is necessary to develop new strategies, such as agitation and activation methods, to improve efficacy.

Studies have shown that calcium phosphate cement incorporated with chitosan and GO (CPC–chitosan–GO), Ag–GO particles and GO–ICG–PDT could effectively reduce the biovolumes of *Enterococcus faecalis* (Figure 1), which has been demonstrated to be one of the most recovered bacteria from the root canals after failed RCT, due to its resistance to the medicament and filling materials [94,110,111,112]. Furthermore, CPC–chitosan–GO excellently supported human dental pulp stem cell (hDPSC) viability, attachment, and growth, with the percentages of live cells at approximately 90%. Ag–GO treatment caused a significant reduction in total biovolumes of multispecies biofilms, including *E. faecalis*, compared to 1% sodium hypochlorite, 2% chlorhexidine, and 17% ethylenediaminetetraacetic acid [111]. These findings indicate that CPC–chitosan–GO paste and Ag–GO are promising for dental applications as root canal sealers and root canal rinses, respectively, to control infections [110]. Recently, PDT has also been used to achieve effective root canal disinfection [112]. Akbari T et al. [112] fabricated a new photosensitizer by incorporating ICG into GO (GO–ICG) and subsequently assessed the antibacterial ability of GO–ICG–PDT against *E. faecalis*. At an energy density of 31.2 J/cm^2^, GO–ICG–PDT significantly reduced the count of *E. faecalis*. Furthermore, GO–ICG–PDT remarkably suppressed the biofilm formation of *E. faecalis* by as much as 99.4%. Therefore, GO–ICG–PDT appears to be viable as a new adjuvant treatment to control endodontic infections.

Pulp regeneration, a promising technique to treat pathological or iatrogenic dental pulp lesions caused by caries or pulpectomy, is expected to replace RCT in the future [113]. In a pilot clinical study, Nakashima M et al. [114] achieved root pulp regeneration through the transplantation of hDPSCs. An ideal pulp sealing restorative material that directly contacts pulp cells should have excellent mechanical behaviors, stimulate odontoblast differentiation, and simultaneously suppress bacterial colonization, none of which have been developed thus far. It is worth noting that in addition to enhancing the mechanical properties and antibacterial activity of materials, increasing evidence has suggested that GO could stimulate the differentiation of hDPSCs into odontoblasts. A modified Ti-based material through a microarc oxidation technique and self-assembled GO (Ti–MAO–GO) has been elucidated as a typical example [115]. Ti–MAO–GO (1.0 mg/mL) dramatically promoted the adhesion, proliferation, and odontogenic differentiation of hDPSCs and exhibited excellent antibacterial activity. The multilayer porous structure of the MAO coating and the physicochemical properties of GO may result in effective direct dentin-like mineralization contact between the pulp and the surface of the pulp sealing material [115]. Incorporation of GO into mesoporous bioactive glass nanoparticle composites could also promote the differentiation of hDPSCs into odontoblast-like cells and potentially induce dentin formation [116]. The expression of genes related to differentiation processes toward phenotypes secreting mineral deposits (e.g., DMP-1 and DSPP) was upregulated by GO-based substrate treatment in DPSCs [42]. Taken together, GO functionalization was proven to induce desired biological effects on DSPCs. However, further study should be conducted to determine the GO concentration for the best compromise of material biocompatibility and effectiveness. For example, the cortical membrane (Lamina) decorated with GO could promote DPSC adhesion, growth, and osteogenic differentiation [117]. Nevertheless, 10 µg/mL GO significantly reduced the cytotoxicity level during 28 days of culture, while 5 µg/mL GO slightly increased the cytotoxicity and reached that of the bare Laminas at 14 and 28 days of culture [117]. In addition, the effects of graphene and other derivatives on the differentiation of DPSCs and whether this differentiation is odontogenic or osteogenic remain to be elucidated.

### 4.3. Suppressing Periodontal Bacteria and Facilitating Tissue Regeneration

Periodontitis is a chronic infectious disease of tooth supporting tissues, including gingiva, cementum, alveolar bone, and the periodontal ligament. Peri-implant diseases are inflammatory conditions that influence the soft and hard around dental implants. They are considered a special subtype of periodontal diseases, which could result in failures of dental implants. The primary goal of periodontitis and peri-implantitis treatment is to control infection and to prevent tissue destruction by removing multispecies biofilms on the root/implant surface and reducing tissue invasion.

*Porphyromonas gingivalis* is regarded as a keystone periodontal pathogen, so it is the target bacteria for many new dental drugs and materials development research. GO nanosheets could effectively suppress the viability of *P. gingivalis* in a dose-dependent manner, stopping growth at a concentration of 40 μg/mL [96]. The mechanisms are involved in destroying the cell wall and membrane, thereby resulting in plasma leakage [96]. GO nanosheets could also inhibit the growth of *Fusobacterium. nucleatum* via a similar mechanism [96]. Thus, whether the inhibitory effect of GO nanosheets on periodontal pathogens is specific needs further investigation. Studies also demonstrated that graphene on a Ti surface could destroy the intact structures of polymicrobial biofilms, including *P. gingivalis*, *F. nucleatum*, and *S. mutans* [118]. The underlying mechanism regarding the bactericidal property might be elucidated as electron transfer from the bacterial biofilms to the graphene-reinforced sample, which disturbed the bacterial respiratory chain and led to a reduction in microbial viability [118]. GQDs coupled with curcumin could effectively suppress the viability of multiple biofilm formation capacity (76%) of peri-pathogens (*Aggregatibacter actinomycetemcomitans*, *P. gingivalis*, and *Prevotella intermedia*). This is associated with ROS generation and the downregulation of biofilm genes (*rcpA*, *fimA*, and *inpA*) [119]. The combined use of brush and a high concentration (≥256 μg/mL) of GO could eliminate residual bacteria and inhibit biofilms consisting of *S. mutans*, *F. nucleatum*, and *P. gingivalis* reformation on implants well, and the effect was significantly better than that of single use [120]. Taken together, these studies indicated that graphene and its derivatives could effectively suppress periodontal infection (Figure 1).

The regeneration of lost tooth/implant-supporting tissues is an ambitious purpose of periodontal regenerative therapy. In stem cell-based therapy of dentistry regeneration, cells and/or growth factors often need to be delivered to the injured site by scaffolds. Periodontal ligament stem cells (PDLSCs) are readily available mesenchymal stem cells (MSCs) with promising applications in regenerative therapy. GO-based materials, such as GO combined with fibroin, GO-coated Ti substrates and GO-applied scaffolds, are regarded as promising biomaterials for tissue engineering due to their biological compatibility and bioactivity to promote the proliferation of PDLSCs [121,122,123]. Specifically, the deposition of an ultrathin film, consisting of alternative deposition of GO and lysozyme (GO/Lys), on the Ti surface could enhance the osteogenic differentiation efficiency of hDPSCs [16]. Thus, antibacterial and osteogenic film functionalization of the implant surfaces may provide new insights for the fabrication of novel implant materials in the future [16]. Moreover, PDLSCs seeded on the GO-coated Ti substrate exhibited a significantly higher proliferation rate and ALP activity as well as upregulated expression of osteogenesis-related genes (COL-I, ALP, BSP, Runx2, and OCN) compared to those on the control Na–Ti substrate [121]. This result suggested that the osteogenic differentiation of PDLSCs on the GO–Ti substrate was higher than that on the Na–Ti substrate. In addition, Nishida E et al. [123] successfully fabricated a GO-modified scaffold by coating the surface of a collagen scaffold with a GO dispersion, presenting improved physical properties, including compressive strength, enzyme resistance, and adsorption of calcium and proteins. Moreover, GO application significantly and dose-dependently increased the proliferation of osteoblastic MC3T3-E1 cells. According to the evaluation of the subcutaneous tissue response in rats, implantation of the scaffold with GO (1 μg/mL) induced remarkable cellular and tissue ingrowth behavior (approximately 2.5-fold greater than that of the collagen scaffold) and prominent ED2-positive macrophage infiltration and blood vessel formation. They further evaluated the effect of GO scaffolds on bone induction in dog tooth extraction sockets and found that following GO scaffold implantation, new bone formation was enhanced fivefold compared to that following control scaffold implantation. The results from this study indicated that GO had good biocompatibility and high bone-induction capability, suggesting that the GO-modified scaffold is expected to facilitate bone tissue regeneration therapy (Figure 3). Qian W et al. [124] fabricated minocycline hydrochloride (MH)-loaded GO films and evaluated their therapeutic effect in beagle dogs using the peri-implantitis model established with silk ligature. The results of radiographic and micro-CT analysis showed that the Ti and MH/Ti groups (especially the Ti group) presented substantial marginal bone loss, with little less bone observed in the GO/Ti group and negligible bone loss observed in the MH/GO/Ti group. The histological analysis showed that the Ti and MH/Ti groups showed many neutrophils, while almost no neutrophils were found in the GO/Ti and MH/GO/Ti groups, in which a large number of osteocytes were observed. Collectively, these results suggested that MH-loaded GO films on implant abutment surfaces exhibited good therapeutic effects for peri-implantitis and could prevent the further development of peri-implantitis in beagle dogs. Collectively, GO-based materials are promising materials to facilitate periodontal tissue regeneration, as summarized in Figure 4.

Several lines of evidence have suggested the involvement of rGO in the proliferation of stem cells [34,125]. For example, rGO-incorporated chitosan nanocomposites could provide a suitable environment for the adhesion and proliferation of human mesenchymal stem cells (hMSCs), enhance cell–substrate interactions and cell–cell contacts, and promote the osteogenesis and neurogenesis of hMSCs [34]. The proliferation rate of hPDLSCs was consistently promoted in certain combinations containing a high dose of silk fibroin and low amounts of GO/rGO. Remarkably, GO/rGO-loaded bilayer composites could induce moderate proliferation and favor osteo/cementoblast differentiation of hPDLSCs without any growth factors. These results suggest the potential applications of rGO in the field of stem-cell-based regenerative therapy in dentistry [125].

### 4.4. Implant Coating and Improving Osseointegration

Dental implants are now a common way to repair permanent tooth loss caused by oral diseases, especially dental hard tissue diseases and periodontitis. Ti is the most commonly used material in implantology, mainly due to its good biological compatibility. However, it has been demonstrated to release alloy particles and ions into peri-implant tissues, which could result in bone loss and osseointegration failure of the implant. Osseointegration is the “gold standard” used to assess the success of implants, and the interactions between the material surface and cells can be regulated by the characteristics of an implant surface. Therefore, considerable efforts have been made to optimize the properties of Ti implant surfaces to improve osteointegration. GO is regarded as a promising candidate to functionalize the surfaces of implants for the regulation of the interactions between implant surfaces and cells, and several groups have explored whether and how GO coatings of dental implants could improve osseointegration [18].

In general, GO coatings can significantly enhance bone marrow mesenchymal stem cell (BMSC) adhesion, spreading, proliferation, and osteogenic differentiation in vitro and promote bone-implant osseointegration in vivo [126]. More new bone mass and fewer gaps between implants and bone tissue were observed around the implants in the GO coating group than in the control group, as shown by the van Gieson (V-G) staining of hard tissue sections [126]. Consistent with this, sequential fluorescence double-labeling showed that almost no alizarin red staining at the SLA implant periphery was observed at 2 weeks, while the SLA/GO group had significant alizarin staining (at 2 weeks) and greater fluorescence intensity (at 4 weeks), indicating the role of GO coating in promoting bone deposition around implants [126]. The underlying mechanisms of these biological effects of GO are associated with upregulating the expression of focal adhesion kinase (FAK) inside cells and its downstream MAPK/P38 signaling pathways, as well as focal adhesion on the GO-modified surface [126]. The above effects can also be enhanced by GO coating modification in different ways, such as DEX–GO–Ti and sandblasted and acid-etched Ti discs with GO [49,127]. The DEX–GO–Ti exhibited a much higher potential to promote rat BMSC proliferation than DEX–rGO–Ti and DEX–control. Similarly, DEX–GO–Ti induced significantly higher alkaline phosphatase (ALP) activities and expression of calcium, osteocalcin (OCN), and osteopontin (OPN) than DEX–rGO–Ti and DEX–control [49].

Interestingly, GO was also studied as an efficient carrier to deliver therapeutic proteins, such as BMP-2 and SP [47]. La W et al. fabricated GO-coated Ti substrates, in which positively and negatively charged GO (GO-NH 3 +/GOCOO−) sheets were assembled layer-by-layer, and then BMP-2 was loaded on the GO-coated Ti substrate via the outermost coating layer of GO-COO−(Ti/GO-). As expected, the GO-coated Ti substrate enabled the bulk loading and sustained release of BMP-2 without influencing the drug structure and bioactivity. BMP-2 delivery using Ti/GO- presented a higher extent of osteogenic differentiation of human bone marrow-derived mesenchymal stem cells in vitro than delivery using pure Ti. Moreover, compared with Ti, Ti/GO-, or Ti/BMP-2 implants, Ti/GO-/BMP-2 implants show more robust new bone formation in mouse models of calvarial defects [47]. Furthermore, they showed that codelivery of SP using Ti or GO-coated Ti further promoted bone formation. Compared with other groups, the dual delivery of BMP-2 and SP (BMP-2/SP/Ti/GO-) presented the greatest new bone formation on Ti implanted in the mouse calvaria. Therefore, delivery of BMP-2 and SP using GO may be useful to improve the osteointegration of Ti in dental implants [48].

Ti implants modified with graphene nanocoating (GN) can maintain their quality and electrochemical and structural integrity under biologically relevant stresses such as microbial-rich environments and inflammatory macrophages. Specifically, Ti implant surfaces with GN exhibited higher polarization resistance and lower corrosion rates after exposure to *S. mutans* supplemented with sucrose for 8 days [128]. GN coverage and structural features were not affected by *Candida albicans* biofilm growth and removal [128]. However, significant loss of this coating was observed as implants were installed and removed from bone, especially in the middle and lower parts of tapered dental implant collars [128]. In addition to being used as a coating, graphene can also be incorporated directly into implants. A study demonstrated that Ti biomaterial containing graphene (Ti-0.125G) could effectively inhibit the viability of multispecies biofilms and exert a high potential to enhance human gingival fibroblast (HGF) viability, adhesion, proliferation, and migration [118]. This may be due in part to the ability of Ti–0.125G to upregulate the expression of genes associated with adhesion (ITGB1, VCL, and FAK) and extracellular matrix components (FN1 and COL1A1) and to activate the FAK signaling pathway in HGF [118]. Furthermore, to mimic the peri-implant environment more rigorously, the bacterial multispecies and HGFs cocultured model was used, and the results revealed that the graphene-reinforced samples could simultaneously benefit HGF responses and suppress bacterial growth, which could enhance the soft tissue integrity and improve its antibacterial infection ability around dental implants [118].

MPCR-TNZ, the multipass caliber-rolled Ti alloy of Ti13Nb13Zr, has been reported to have strong fatigue characteristics and high mechanical strength. For further applications in dental practice, Jung HS et al. obtained DEX/rGO–MPCR-TNZ by modifying the MPCR-TNZ surface with rGO followed by the loading of osteogenic DEX via π-π stacking on the graphitic domain of rGO. They found that DEX/rGO–MPCR-TNZ significantly stimulated MC3T3-E1 cell growth and osteoblast differentiation, showing remarkable expression of osteogenic markers, including Runx2, OPN, OCN and Col-1. rGO–MPCR-TNZ also exhibited an increase in calcium nodule deposition and ALP activity. The prototype implant of rGO–MPCR-TNZ was successfully implanted onto an artificial bone block with mechanical performance and structural characteristics similar to those of a jawbone, and the rGO on the surface remained stable even after implantation, confirming the feasibility of rGO–MPCR-TNZ for applications in clinical dentistry. Shin YC et al. [129] prepared SLA Ti implants with different modifications on the surface, including rhBMP-2 immobilization or treatment and rGO coating, and compared the cellular behaviors in vitro and the osseointegration activity in vivo among different implants (Figure 5A). They found that rGO-coated Ti promoted the growth and osteogenic differentiation of cells significantly even without any osteogenic factors. Moreover, the implantation of rGO-coated Ti significantly promoted the osseointegration of the implants and tissue regeneration in animal models compared with the other three types of Ti implants (Figure 5B) [130].

Taken together, to date, there have been studies using graphene, GO, and rGO as coatings for Ti implants. Evidence from both in vitro and in vivo studies has suggested that GO coatings are beneficial for peri-implant bone formation and stabilization. The rGO coating also showed the potential to promote osteoblast differentiation; however, the pro-osteogenic effect of the rGO coating needs further investigation in vivo. Although graphene coatings have good stability in bacteria-rich and inflammatory environments, the ability of graphene coatings to cope with mechanical wear needs to be improved, and their osseointegration is unclear. Here, we summarize the applications of graphene and its derivatives mentioned above in dentistry for reference by subsequent researchers (Table 1).

## 5. Perspective and Summary

Taken together, graphene and its derivatives can improve the mechanical properties of dental materials and present good biocompatibility; it may be an ideal material for caries filling, especially for severe caries filling. Existing evidence indicates that graphene and GO can inhibit the growth of both *S. mutans* and *P. gingivalis*, which is beneficial for caries and periodontal disease therapy, as well as the success of implantation. However, many materials have this effect, and the current opinion is that dental caries and periodontal disease are not infections caused by a single pathogenic bacterium but result from oral microbiota dysbiosis and the imbalance of oral microbiota and host interactions. If graphene and its derivatives can selectively inhibit harmful bacteria without influencing beneficial bacteria, they will become a modern and powerful weapon for dentistry. Moreover, although graphene and GO can promote the differentiation and proliferation of DPSCs and PDLSCs, which is beneficial to the regeneration of dental pulp and periodontal tissues, whether they can act on specific genes or proteins to precisely regulate the host immune inflammatory response and maintain the balance between oral microbes and the host remains to be studied. In addition, the development of dentistry is inextricably linked to the delivery of biological drugs, including peptides, monoclonal antibodies, and nucleic acids. Graphene and its derivatives have shown great promise in the development of drug delivery systems, especially the delivery of drugs for targeted cancer therapy. However, there are currently only several lines of evidence that GO can be applied to oral biopharmaceutical delivery.

## Figures and Tables

**Figure 1 ijms-23-04737-f001:**
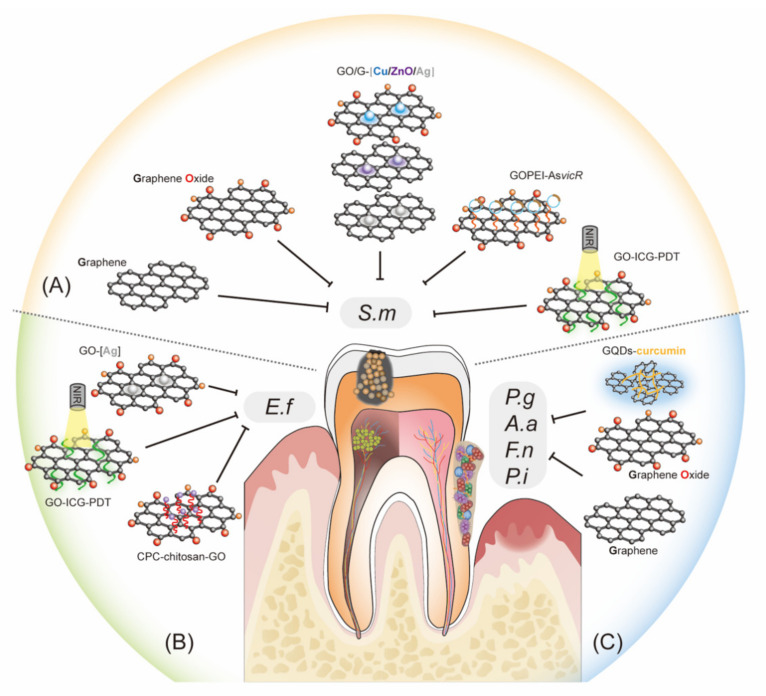
Graphene-based materials can inhibit oral bacteria. (**A**) Inhibiting cariogenic bacteria: graphene (G) and graphene oxide (GO) may directly inhibit *S. mutans* and indirectly suppress *S. mutans* by enhancing the antibacterial properties of metallic nanomaterials such as silver (Ag), copper (Cu), and zinc oxide (ZnO) nanoparticles. GO may also be combined with new technologies to achieve its antibacterial activity against *S. mutans* by delivering nucleic acids and photosensitizers. (**B**) Control dental pulp infection: Ag–GO particles, calcium phosphate cement (CPC) incorporated with chitosan and GO (CPC–chitosan–GO), and antimicrobial photodynamic therapy (PDT) using indocyanine green (ICG) as a photosensitizer delivered by GO (GO–ICG–PDT) can effectively reduce the biovolumes of *E. faecalis*. (**C**) Suppression of periodontal pathogens: GO, G, and graphene quantum dots (GQDs) coupled with curcumin (GQDs–curcumin) may inhibit the growth and multispecies biofilm formation of periodontal pathogens, such as *P. gingivalis*, *A. actinomycetemcomitans*, *F. nucleatum*, and *P. intermedia*. *S.m*, *Streptococcus mutans*; *E.f*, *Enterococcus faecalis*; *P.g*, *Porphyromonas gingivalis*; *F.n*, *Fusobacterium nucleatum*; *A.a*, *Aggregatibacter actinomycetemcomitans*; *P.i*, *Prevotella intermedia*; PEI, *polyethylenimine*; AsvicR, Antisense vicR RNA.

**Figure 2 ijms-23-04737-f002:**
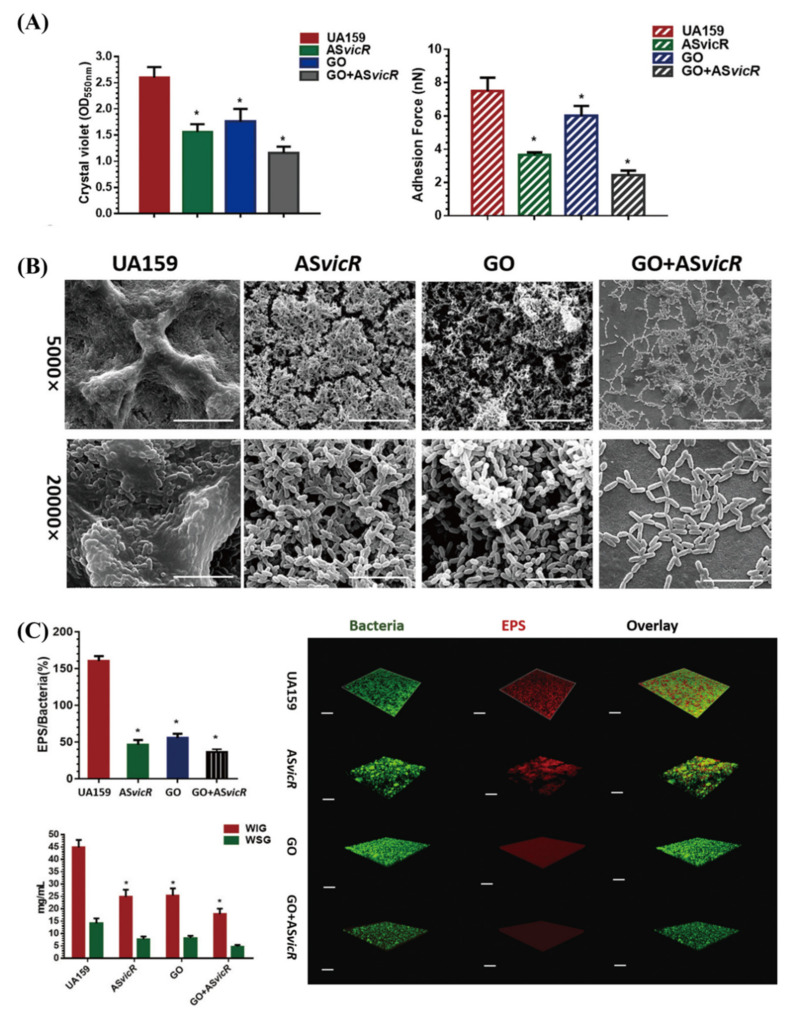
Nanographene oxide (GO) with antisense vicR (AsvicR) RNA inhibited the biofilm formation of *Streptococcus mutans*. (**A**) Biomass and the values of adhesion force of *S. mutans* biofilms; (**B**) SEM of *S. mutans* biofilms, scale bar for 5000×/20,000× magnification, 20 μm/5 μm; (**C**) volume ratio of the exopolysaccharide (EPS) matrix to the bacterial biomass in *S. mutans* biofilms, green for bacteria and red for the EPS matrix; scale bars, 100 μm. * *p* < 0.05. [104] Copyright 2020, The Japanese Society for Dental Materials and Devices.

**Figure 3 ijms-23-04737-f003:**
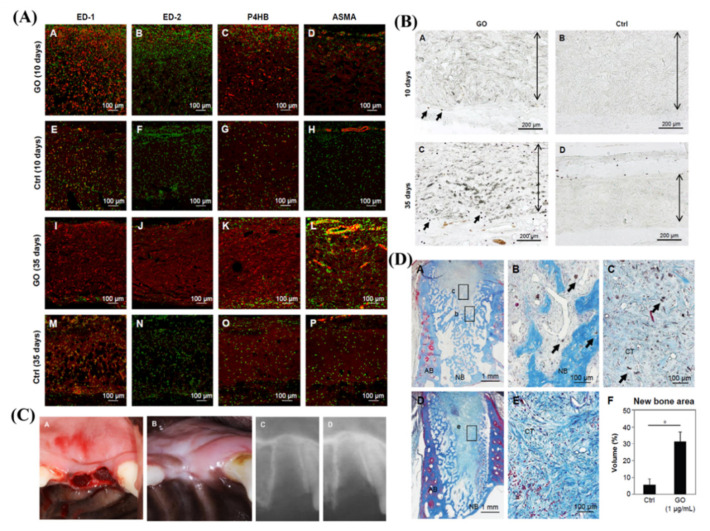
**Graphene oxide (GO) coating on a collagen scaffold enhances alveolar bone regeneration.** (**A**) Immunohistochemical assessment of subcutaneous tissue with scaffold implantation. ED1 and ED2 indicate macrophages, P4HB indicates fibroblasts, and ASMA indicates blood vessels, in red. I (**B**) Peroxidase staining of subcutaneous tissue implanted with scaffold. Arrows and double arrows indicate peroxidase-positive granulocytes and the implanted scaffold, respectively. (**C**) Photographs and radiographic images immediately and 2 weeks after the implantation of the GO scaffold (**D**) Histological assessments of new bone formation in the extraction socket at 2 weeks after scaffold implantation; arrows indicate residual GO. *, *p* < 0.05. [123] Copyright 2016, the Author(s).

**Figure 4 ijms-23-04737-f004:**
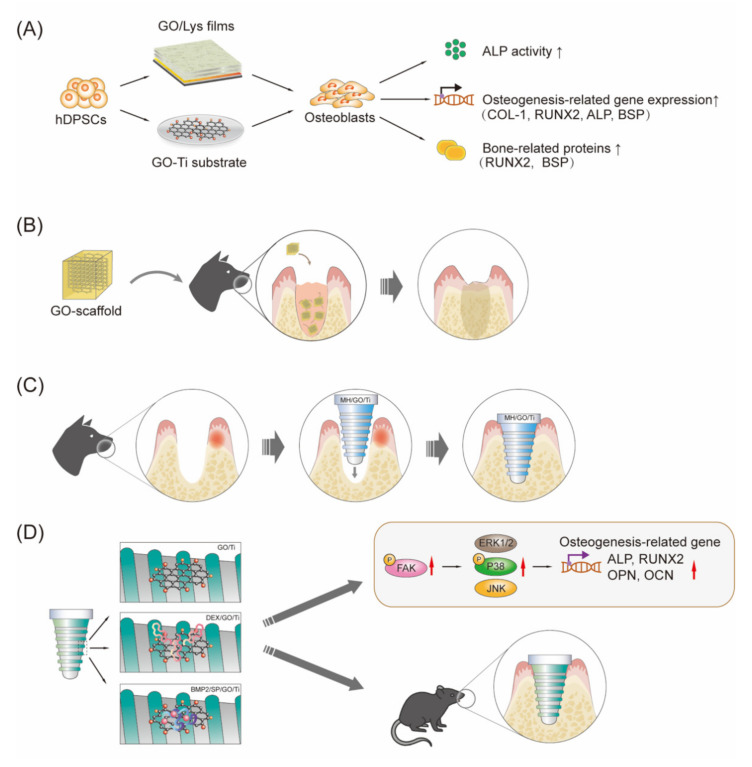
Graphene-based materials possess great potential to facilitate bone regeneration in dentistry. (**A**) Graphene-based materials (e.g., GO and lysozyme (GO/Lys)8 films on Ti substrate, GO coated Ti substrate) induce PDLSC proliferation and osteogenic differentiation in vitro. (**B**) Implantation of the GO–3D collogen sponge-form scaffold (GO–scaffold) in the tooth extraction socket enhances the formation of new bone. (**C**) Ti implants with MH-loaded GO films (MH/GO/Ti) show good therapeutic effects for peri-implantitis and can prevent its further development in beagle dogs, exhibiting a significantly reduced modified sulcus bleeding index, marginal bone loss and peri-implant probing pocket depth, and many osteocytes but almost no neutrophils. (**D**) GO coating on the Ti surface facilitates bone marrow-derived mesenchymal stem cell osteogenic differentiation by activating intracellular FAK and its downstream MAPK/P38 signaling pathway to upregulate the expression of genes associated with osteogenesis (such as ALP, OPN, RUNX2 and OCN) and enhance bone-implant osseointegration in vivo. The loading of dexamethasone (DEX) (DEX/GO/Ti), bone morphogenetic protein 2 and/or substance P (BMP-2/SP/GO/Ti) can enhance the above effects. The black arrow indicates promote; the red arrow indicates upregulation. PDLSCs, periodontal ligament stem cells; MH, minocycline hydrochloride; FAK, focal adhesion kinase.

**Figure 5 ijms-23-04737-f005:**
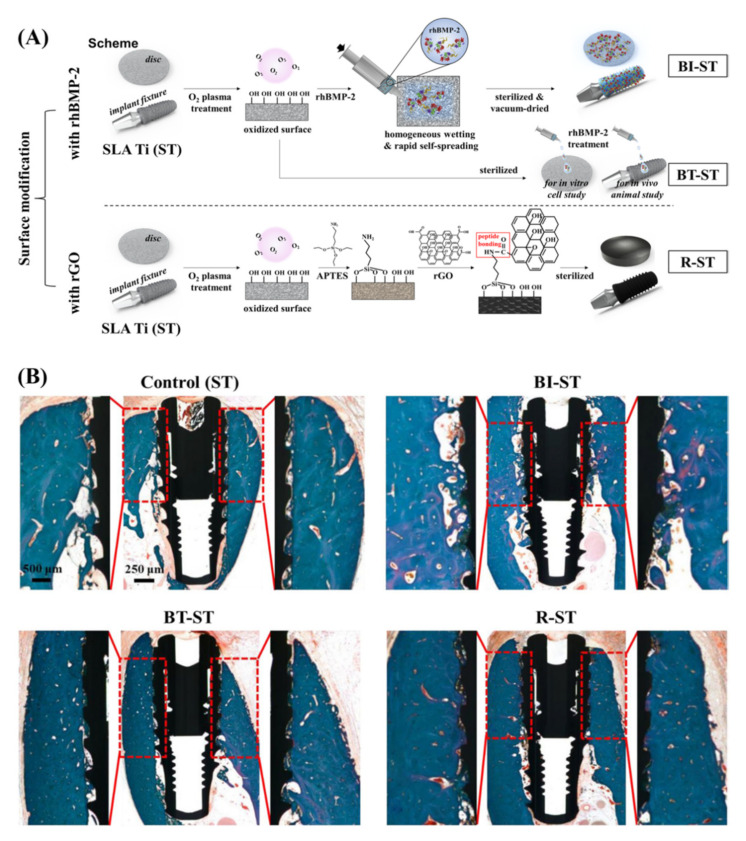
A reduced graphene oxide (rGO) coating enhanced the osseointegration of dental implants. (**A**) Scheme for surface modification of SLA Ti (ST) with rhBMP-2 immobilization (BI-ST) or treatment (BT-ST) and rGO coating (R-ST). (**B**) Histological assessment for osseointegration of implants via Goldner Trichrome staining; the region of interest for measurements of bone-to-implant contact length and intra-thread bone density is exhibited with higher magnification. [130] Copyright 2022, the Author(s).

**Table 1 ijms-23-04737-t001:** Summary of graphene and its derivatives in dentistry.

Main Subject	Form of Graphene Materials	Method	Material Type	Role and Advantages	Ref
Dental Materials (Restorative Dentistry)	Graphene–Ag nanoparticles (G–AgNp) Graphene oxide (GO)	Adding G–AgNp to a PMMA auto-polymerizing resin GO sheets were infused into primer	PMMA auto-polymerizing resin Primer	Antibacterial activity, minimal toxicity, improved flexural properties. Enhance shear bond strength	[68] [88]
Endodontics	Graphene Oxide (GO) Graphene oxide (GO)	Nano-graphene oxide with antisense vicR RNA plasmid (GO–PEI–ASvicR). Graphene oxide (GO) adhesive	Plasmid Adhesive resin	Antibacterial (*S. mutans*) Shows comparable bond strength and durability of resin dentine bond.	[104] [87]
Periodontics	Graphene	Graphene Quantum Dot coupled with curcumin (GQD–Cur)	Photosensitizing agents	Downregulation of the biofilm genes expression	[119]
Implantology	Graphene oxide (GO) Graphene Reduced graphene oxide	Graphene oxide (GO) deposition (on a zirconia surface) Mg alloy with graphene nanoparticles (Gr) Reduced graphene oxide (rGO)-coated sandblasted	Direct-deposited graphene oxide on dental implants Coated on dental implants Coated on dental implants	Inhibited the attachment of *S. mutans* and stimulated proliferation and differentiation of osteoblasts. High cytocompatibility and superior osteogenic properties Accelerate the healing rate with the high potential of osseointegration.	[95] [88] [130]
Tissue engineering	Graphene oxide (GO)	GO dental materials	A rat model of a non-critical mandibular defect.	Bone regeneration and biocompatibility	[70]

## Data Availability

No new data were created or analyzed in this study. Data sharing is not applicable to this article.

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
