# Peer review of "Research on Graphene and Its Derivatives in Oral Disease Treatment"

_ijms, 2022, doi:10.3390/ijms23094737_

Round 1

Reviewer 1 Report

You can find attached the PDF file.

Author Response

The aim of this review was to evaluate the development of Graphene and its derivatives in oral diseases treatment. The paper is of interest; however, some criticisms should be addressed before publication.

  1. Line 67: the term “slow” must be check. Is the term ”low” more appropriated?

      Reply: Thanks. We think “low” is the more appropriated term here.

  1. Line 102: “SP2”. Must be explained.

      Reply: We have explained it in the revised manuscript.

  1. Line 158: “SN2”. Must be explained.

      Reply: We have explained it in the revised manuscript.

  1. Line 186: “at a concentration 4 g/mL,”. This sentence must be checked.

      Reply: Thanks, we have checked it and revised it in the manuscript.

  1. Line 191: The acronym DPSCs must be explained.

      Reply: We have explained it in the revised manuscript.

  1. Line 259: One of the term ”the” must deleted.

      Reply:  We have deleted the term “the”.

  1. Line 288: “there is no natable difference”. This sentence must be checked.

      Reply:  We have checked it and revised it in the manuscript.

  1. Line 325: The acronym MTT must be explained.

      Reply:  We have explained it in the revised manuscript.

  1. Line 385: Reference must be written between squared brackets, and not in exponent.

      Reply:  Thanks, we have revised the citation format of references.

  1. Line 492: A final point must be deleted.

      Reply:  we have deleted it. 

  1. Line 541: Reference must be written between squared brackets, and not in exponent.

      Reply:  Thanks, we have revised the citation format of references.

  1. Line 541: “*P,0.05; Ctrl, control.” Is this information necessary? The star is not linked to another one.

      Reply:  Thanks, we have deleted this information.

  1. Line 585: The acronym V-G must be explained.

      Reply:  We have explained it in the revised manuscript.

  1. Line 648: Figure must be referenced between parentheses instead squared brackets.

            Reply:  We have revised it.

  1. Line 652: Figure must be referenced between parentheses instead squared brackets.

      Reply:  We have revised it.

  1. Line 659: Reference must be written between squared brackets, and not in exponent.

      Reply:  Thanks, we have revised the citation format of references.

Reviewer 2 Report

Authors provided a review of the development and applications of graphene nanomaterials in dentistry. Graphene nanomaterials demonstrate outstanding characteristics, including good biological compatibility and the ability to inhibit the growth of bacteria associated with periodontal disease. The authors performed a systematic review of the field and provided valuable insights into the perspectives. However, the overall quality of the draft could be improved if some blemishes are fixed.

1. The first two paragraphs in the introduction are too wordy. It will be better if the contents can be summarized in a more concise way. And if the overall structure of the paper can be summarized at the end of the introduction section will provide a better view for the audiences.

2. The citation should be added to where the studies are mentioned instead of always putting at the end of the sentences.

3. Some journal names of the reference are not fully provided. For example, "Beilstein Journal of Nanotechnology" should be used instead of "Beilstein J Nanotechnol". 

Author Response

Authors provided a review of the development and applications of graphene nanomaterials in dentistry. Graphene nanomaterials demonstrate outstanding characteristics, including good biological compatibility and the ability to inhibit the growth of bacteria associated with periodontal disease. The authors performed a systematic review of the field and provided valuable insights into the perspectives. However, the overall quality of the draft could be improved if some blemishes are fixed.

  1. The first two paragraphs in the introduction are too wordy. It will be better if the contents can be summarized in a more concise way. And if the overall structure of the paper can be summarized at the end of the introduction section will provide a better view for the audiences.

      Reply:  Thanks for your suggestion. We have rewritten the first two paragraphs to make it  

      more concise and summarized the overall structure of the paper at the introduction section.

  1. The citation should be added to where the studies are mentioned instead of always putting at the end of the sentences.

Reply:  We have revised the position of part of the citations.

  1. Some journal names of the reference are not fully provided. For example, "Beilstein Journal of Nanotechnology" should be used instead of "Beilstein J Nanotechnol". 

Reply:  We have checked this problem and revised it.